# Feasibility of motor imagery and effects of activating and relaxing practice on autonomic functions in healthy young adults: A randomised, controlled, assessor-blinded, pilot trial

**Turhan Kahraman**[1], **Derya Ozer Kaya**[1], **Tayfun Isik**[1], **Sukriye Cansu Gultekin**[1], **Barbara Seebacher**[2,3] *

**1** Department of Physiotherapy and Rehabilitation, Faculty of Health Sciences, Izmir Katip Celebi University, İzmir, Turkey, **2** Clinical Department of Neurology, Medical University of Innsbruck, Innsbruck, Austria, **3** Department of Rehabilitation Research, Rehab Centre Muenster, Muenster, Austria

* barbara.seebacher@i-med.ac.at

## Abstract

### Introduction

Motor imagery (MI) is the mental rehearsal of a motor task. Between real and imagined movements, a functional equivalence has been described regarding timing and brain activation. The primary study aim was to investigate the feasibility of MI training focusing on the autonomic function in healthy young people. Further aims were to evaluate participants' MI abilities and compare preliminary effects of activating and relaxing MI on autonomic function and against controls.

### Methods

A single-blinded randomised controlled pilot trial was performed. Participants were randomised to the activating MI (1), relaxing MI (2), or control (3) group. Following a MI familiarisation, they practiced home-based kinaesthetic MI for 17 minutes, 5 times/week for 2 weeks. Participants were called once for support. The primary outcome was the feasibility of a full-scale randomised controlled trial using predefined criteria. Secondary outcomes were participants' MI ability using the Movement Imagery Questionnaire-Revised, mental chronometry tests, hand laterality judgement and semi-structured interviews, autonomic function.

### Results

A total of 35 participants completed the study. The feasibility of a larger study was confirmed, despite 35% attrition related to the COVID-19 pandemic. Excellent MI capabilities were seen in participants, and significant correlations between MI ability measures. Interview results showed that participants accepted or liked both interventions. Seven major themes and insider recommendations for MI interventions emerged. No significant

**Data Availability Statement:** All relevant data are within the paper and its Supporting Information files.

**Funding:** The assessment devices and physiotherapy laboratories used in this study was funded by the Department of Scientific Research Projects, Izmir Katip Celebi University (Project ID: 2017-2ÖNP-SABF-0008, Title: Establishment of Physiotherapy and Rehabilitation Vocational Skills, Education, Research, and Innovation Laboratory Infrastructure); https://www.ikcu.edu.tr/ The funders had no role in study design, data collection and analysis, decision to publish, or preparation of the manuscript.

**Competing interests:** The authors have declared that no competing interests exist.

differences and negligible to medium effects were observed in MI ability or autonomic function between baseline and post-intervention measures or between groups.

## Conclusions

Results showed that neither activating nor relaxing MI seems to change autonomic function in healthy individuals. Further adequately powered studies are required to answer open questions remaining from this study. Future studies should investigate effects of different MI types over a longer period, to rule out habituation and assess autonomic function at several time points and simultaneously with MI.

## Introduction

Most current physiotherapy approaches are based on the training of real movements to stimulate damaged motor neural connections through neuroplasticity-related mechanisms [1]. As a complementary strategy, motor imagery (MI) has increasingly been used in the physiotherapy field, defined as a mental rehearsal of movement without revealing a real movement [2]. MI is frequently categorised into visual and kinaesthetic imagery [3]. Kinaesthetic imagery would mean that an individual "senses" his/her own body moving. In contrast, visual imagery would be referred to when an individual "watches" the execution of an imagined motor action. Various studies have reported an activation of similar brain areas during movement execution and MI [4–6] and a similar duration of imagined and real movements (temporal congruence) [7]. Both the overlap in brain areas and comparable timing of real and imagined movements are referred to as their functional equivalence [8].

Several studies have found that MI training improves balance and mobility in older adults [9]. Accelerated functional recovery [10] and upper limb motor function gains have been observed after MI in individuals post stroke [11]. MI improves muscle strength in healthy adults [12] and facilitates motor function in people with Parkinson's disease [13]. Moreover, RCTs have shown that MI is effective for people with multiple sclerosis [14–18].

Numerous studies have investigated the mechanisms of MI [5, 6, 19, 20]. Some have suggested MI training-related benefits induced by improved motor learning, due to a strengthening of synaptic connections [6, 20]. Others have proposed changes in metabolic responses caused by MI training, comparable to those following real exercise training. A pioneering study observed similar acute effects on physiological parameters such as heart rate, oxygen consumption, blood pressure, respiratory rate, and metabolic rate after MI practice as after actual exercise training in healthy college students [21]. However, there is a lack of evidence about the effects of MI training over a prolonged time on autonomic functions [22]. The primary aim was to investigate the feasibility of MI training focusing on autonomic function in healthy young individuals. Further aims were to assess participants' MI abilities and compare the preliminary effects of activating and relaxing MI on autonomic function against a control group.

Study hypotheses were that a full-scale RCT would be feasible, healthy young individuals would show high MI abilities, moderate to high correlations between MI ability measures would be seen, and trends for a change in autonomic function would be observed from baseline to post-intervention, with contrasting effects after activating and relaxing MI, as compared to the control group.

## Materials and methods

### Study design and setting

This pilot RCT was designed as an assessor-blinded trial with three parallel groups (1:1:1 allocation ratio) and carried out from 21.11.2019 to 31.3.2020. Assessments were performed at the Physiotherapy and Rehabilitation Laboratory, Izmir Katip Celebi University, Izmir, Turkey. Interventions were home-based and participants were supported by pre-recorded audio-visual files containing instructions and MI exercises.

The study protocol followed the CONSORT statement (S1 Checklist), was prospectively registered with ClinicalTrials.gov (NCT04171271), approved by Non-invasive Research Ethics Board of Izmir Katip Celebi University (reference 2019/399, 26.09.2019) and performed following the ethical standards of the Declaration of Helsinki. All participants provided written informed consent before they participated (see S1 Study protocol).

### Participants

Healthy young individuals were recruited. This study was advertised at six classes (362 students) at Izmir Katip Celebi University, Izmir, Turkey. Inclusion criteria were: >18 years of age, apparently healthy and absence of any neurological or orthopaedic disease. Exclusion criteria were pregnancy and history of any previous disorder or surgery, altering the physical performance or physiological functions.

Eligible students interested were randomised into one of three groups: activating kinaesthetic MI training (group 1), relaxing kinaesthetic MI training (group 2), and controls (group 3).

Twelve participants per group are considered a minimum sample size [23], and 10% of the total sample size of a full-scale study is suggested for pilot studies [24]. Based on a power of 80%, a two-tailed alpha of 0.05, mean 1 of -2.13, mean 2 of 0.20 and pooled standard deviation of 4.67 from the group differences in respiratory rates of an MI training study that used physiological measures [21], a sample size of 189 participants for three groups resulted (http://powerandsamplesize.com/). Considering these recommendations, the sample size was determined as 15 participants per group (by addition of 20% attrition, a total of 54 participants).

Participants of this study were invited for semi-structured interviews, asking questions regarding their experiences of the MI and the acceptability of the intervention. The number of interviewees was based on the concept of saturation. Data saturation is reached when no additional information (themes) is expected from further interviews, sufficient information has been gathered to replicate the study and further coding is no longer viable [25]. Guest et al. have found that this was achieved with six to twelve interviews [26].

### Randomisation, allocation concealment and blinding

Recruitment used convenience sampling and simple randomisation with allocation concealment. The randomisation and assignment of participants to groups were performed by a researcher who was not involved in the assessment or intervention procedures (BS). Randomisation was based on a single sequence of random assignments [27] using a random number list (https://www.randomizer.org/). Participants were enrolled by the study PI (TK). Assessors were blinded to the group allocation until study completion. Participants were asked not to disclose their group allocation to anyone until the study was completed.

### Study intervention

In group 1, the intervention consisted of activating kinaesthetic MI training. In group 2, the intervention comprised relaxing kinaesthetic MI training. In both intervention groups, due to

the piloting character, the duration was two weeks. Participants were asked to practice MI at home for 17 minutes per day and 5 times per week. The practice duration and frequency were chosen according to a previous study [28]. An MI script was written for the interventions. The video footage of the exercises used for MI tasks was created together with MI instructions. The PETTLEP [8] ideas were applied to support participants' understanding of the intervention in their respective groups and their effectiveness (Position/Physical, Environment, Tasks, Timing, Learning, Emotion and Perspective).

Following the allocation and baseline assessments, participants in the intervention groups received one session of an MI introduction programme comprising MI theory and practice [29]. Additionally, participants were carefully instructed about their suggested physical positions during MI practice, choice of a suitable, quiet training environment, the tasks to be performed, the timing of imagined movements, how to enhance their MI experience, arousal related (group 1) or calmness and relaxation related (group 2) and motivational aspects of the MI tasks and perspective in their respective groups [8]. The instructor (TK) and creator (BS) of two audio-visual files, which included the respective exercises created specifically for this study, are physiotherapists with a profound knowledge and practical experience in MI. Training files could be accessed from a password-protected cloud platform via a shared link. The MI intervention was performed at home and the practice frequency was recorded in an exercise diary (see S1 File for intervention details).

The control group (group 3) received no specific intervention and was only assessed at baseline and 2 weeks later. At the end of week 1, intervention group members were called for support with their MI and as a reminder of the post-intervention assessment. For equivalency reasons and as a reminder of the second assessment, control group members were also called after week 1.

## Data collection: Demographic measures and body composition

Age, gender, height, and weight of the participants were recorded. Body composition was evaluated using a body composition analyser (MC-780MA, TANITA Corporation of America, Inc. IL). Quantities of body fat, muscle, and water were reported in kg and percent. Additionally, using bioelectrical impedance analyses, the level of visceral fat was reported and ranged from 0 to 10, where higher scores indicated more visceral fat [30]. Adequate validity and reliability of the TANITA system have been demonstrated in college students [31]. The hand dominance (laterality) in everyday activities was assessed using the validated Turkish version of the Edinburgh Handedness Inventory (EHI) [32–34]. Blinded assessors (two licensed physiotherapists) collected data at baseline and post-intervention.

## Primary outcome

**Feasibility.** The primary outcome was the feasibility of conducting a full-scale RCT. The criteria for feasibility success were a target recruitment rate of 10% (or 10 participants per month), a target retention rate of 80% and a target minimum adherence rate of 70% of the overall practice sessions (or 7 MI sessions out of 10 sessions). In addition, high acceptability of the intervention and no serious adverse events were expected. Participants' adherence with and acceptability of the intervention were reported narratively. Both were recorded during phone calls at week 1 and at post-intervention assessments. A record sheet was prepared for any adverse events observed during the intervention sessions and tests, including information about the severity, duration, causality (i.e., whether they were attributable to the intervention), time and date, and clinical action taken. Raw count and percentage of missing data were also recorded.

### Secondary outcomes

**Motor imagery ability.** Four different MI ability assessments were employed because using at least two different approaches is recommended [35, 36].

*Movement Imagery Questionnaire-Revised (MIQ-R).* It assesses the visual and kinaesthetic MI ability and comprises 4 visual and 4 kinaesthetic items [37]. Each item involves performing a movement, followed by imaging that same movement using a visual or kinaesthetic mode. Participants are then asked to rate the ease or difficulty of generating that image on a 7-point Likert scale. Higher scores indicate greater visual or kinaesthetic MI ability. A cut-off point of 25 out of the maximum score of 56 has been suggested previously [38] and hence for each of the two subscales, a 12.5 cut-off point was used to signify good or poor MI ability. Acceptable validity and reliability of the Turkish version of the MIQ-R have been demonstrated [39].

*Mental chronometry (MC) tests.* They assess the temporal equivalence of actual and imagined movements [40]. MC was evaluated employing a walking and writing task [41]. For the walking task, participants imagined and executed walking a 6-metre distance at a comfortable speed. For the writing task, participants imagined and executed writing the following sentence in Turkish: "Türkiye'nin başkenti Ankara" meaning "Ankara is the capital of Turkey." For time taking, participants kept an electronic stopwatch in their non-dominant hand. MI quality was evaluated using the formula 1-((actual movement duration-MI duration)/actual movement duration) [42], where ideal values would be close to 1.

*Hand Laterality Judgement (HLJ).* It was measured using the Recognise™ App (Neuro Orthopaedic Institute, NOI, Adelaide) on a tablet. This App has been found to be valid and reliable in healthy individuals, and minimum detectable changes are available [43, 44]. The accuracy and reaction time for the right and left sides were reported. High values in accuracy and low values in reaction time indicate high MI ability [45]. A meta-analysis has shown values of 2.08 (95% confidence interval (CI): 2.02, 2.15) seconds and 91.9% (95%CI: 91.3%, 92.5%) for the HLJ time and accuracy in healthy controls, respectively [46].

*Motor imagery experiences and acceptability of the intervention.* It was considered important that participants accepted the MI practice as otherwise, one could not expect people to maintain MI training for a prolonged period of time. So, at post-intervention, participants in the intervention groups were invited for semi-structured face-to-face interviews by the study PI. They were asked about their perceptions of the MI interventions, their experiences during the MI and their suggestions for a future MI intervention [47]. Interviews were designed according to recommendations on qualitative interviewing [48] (S2 Checklist) and conducted by two Master's students in physiotherapy (SCG and TI, female and male). Details are presented in S2 File and S1 Table.

**Autonomic functions.** Physiological cardiorespiratory and metabolic responses as indicators of autonomic function were assessed. The basal metabolic rate (BMR) was measured using a cardiopulmonary exercise test device (Cosmed Quark CPET, Cosmed, Rome, Italy). Participants were asked to refrain from eating, caffeine intake or smoking ≥4 hours before the BMR assessment. They were also instructed to avoid strenuous exercise 24 hours in advance of the assessment [49]. Measurements were performed with participants wearing comfortable clothing and lying supine on a bed. The room was ventilated for at least 1 hour before the measurement. Initially, the system was calibrated to allow for accurate $O_2$ and $CO_2$ gas concentration measurement. During the measurement, participants wore a mask designed to cover the mouth and nose; they were asked to lie quietly for 15 minutes. The system automatically recorded data related to the BMR. Reported variables were the resting metabolic rate (RMR) representing the daily caloric expenditure (kcal/day) [50], respiratory quotient (RQ), $O_2$ uptake ($VO_2$) (ml/min), $CO_2$ production ($VCO_2$) (ml/min), minute ventilation (VE) (l/min),

concentrations of $O_2$ (FEO$_2$, %) and $CO_2$ (FECO$_2$, %) in the exhaled gases and respiratory frequency (RF). The RQ is the volume of the $CO_2$ released over the volume of the $O_2$ absorbed during respiration and indicates which macronutrients are being metabolised, as different energy pathways are used for fats (RQ = 0.7), proteins (RQ = 0.8), and carbohydrates (RQ = 1.0). Fat and carbohydrate substrates (%) used for the energy metabolism were also assessed.

## Data analyses

**Statistical analysis.**   IBM SPSS Statistics (Version 26.0. Armonk, NY: IBM Corp) or GraphPad Prism (Version 8, La Jolla, CA) was used for all statistical analyses. Statistical significance was defined as a two-tailed p<0.05. Intention-to-treat analysis was performed for all cases with complete follow-up data, which were analysed by original assigned groups. Descriptive statistics were reported for all outcomes. Continuous data were checked for outliers and normality using the Shapiro-Wilk test, histograms and Q-Q plots. Age, 6-metre walk MC, right-left discrimination judgements, RQ, VCO$_2$, FEO$_2$ and most of the TANITA body composition measures did not satisfy the normality criteria. A reciprocal transformation (1/x) changed the distributions into normal. Due to the small sample size, means (95% confidence interval) and medians (25$^{th}$-75$^{th}$ percentiles) were reported for continuous and ordinal data, respectively and raw count (frequency, percentage) for nominal data.

Correlational analyses were performed between the different MI ability measures to evaluate their relationship. Spearman's rank correlation coefficients of 0.3–0.49 were considered low, 0.5–0.69 moderate, and ≥0.7 strong [51]; they were calculated with their 95%CI and p-values corrected for multiple comparisons using a Bonferroni correction.

The recruitment rate was estimated by dividing the number of participants who consented by the number of people eligible, multiplied by 100. The retention rate was determined: (N who completed the study, divided by N of the total sample) ×100. The adherence rate was reported as the percentage of the planned MI practice sessions (5×/week) actually performed by the participants over the 2-week study period [52]. The recruitment, retention, and adherence rates were calculated with their 95% CI [53]; when the proportion was close to 0 or 1, a Poisson approximation was employed [54]. Little's test was performed to evaluate whether the data were missing completely at random (MCAR). A p>0.05 would signify missingness completely at random i.e., independence of missing values from both the observed and the unobserved data [55] so that a listwise deletion of cases with missing values was safe [56]. Patterns of missing data were analysed (without performing multiple imputations) to identify the missing data percentage.

Based on the CONSORT statement, all statistical tests for baseline comparison between the groups are to be avoided and therefore, data were presented without p-values. For continuous data, two-factor mixed analysis of variance (ANOVA) was employed, with groups as between-subjects factor and time (baseline, post-intervention) as within-subject factor. ANOVA effect size measures were estimated as partial eta squared values ($\eta_p^2$). For all relevant analyses, significant violations of ANOVA were tested for and where appropriate, standard correction procedures were applied. For ordinal data, Kruskal–Wallis test was used across all groups (Gps 1–3) and time points (baseline, post-intervention), followed by Dunn's multiple comparisons test. Kruskal-Wallis effect size measures were calculated as eta squared values ($\eta_H^2 = (H-k + 1) / (n-k)$, where H is the Kruskal-Wallis H statistic, k is the number of groups and n is the total number of observations [57]. Partial eta squared effect sizes of 0.02 are regarded small, 0.13 medium and 0.26 large, and eta squared effect sizes of 0.01 are referred to as small, 0.06 medium and 0.14 large [58].

**Qualitative data analysis.** Interviews were transcribed and analysed manually using Qualitative Content Analysis [59, 60]. Both a data-driven and a concept-driven approach were chosen for the analysis. Details are presented in S2 File. Throughout the analyses, rigour, credibility and reliability were maximised [61, 62] by following a systematic and consistent approach (S1 Checklist). In addition, the entire dataset was double coded by two researchers within two weeks after the initial coding (SCG, TK). After translation into English, the coding was checked by a third researcher (BS) and any discrepancies were solved. Finally, researchers were aware of their effect on the interview process and outcomes based on the concept of reflexivity [63].

## Results

Recruitment was completed as planned. Due to the COVID-19 pandemic-related lockdown, 19 out of 54 participants (35.0%) dropped out of the study (Fig 1). There were 28 females and 7 males with a mean age of 20.3 years (Table 1).

Participants' MI ability and autonomic (metabolic and cardiorespiratory) function at baseline are shown in Table 2.

Of eligible 359 students, 54 were enrolled in the study within 5 months, the recruitment rate was 14.9% (95%CI: 11.5, 19.1%). That is, per month, 10–11 participants could be recruited into the study. Prior to the COVID-19 outbreak, which caused an attrition rate of 35.0%, 100.0% of the participants remained in the study. Considering the 100.0% retention rate observed before the pandemic onset, it is likely that the 80.0% target retention rate would have been achieved. Participants reported the intervention as acceptable or even pleasurable (66.0%). They reported having practiced a median 10 (10.0–10.0 25th-75th percentile) times within the 2-week intervention period, thus, adherence was excellent. There were no reported adverse events related to this study. Little's test of missing completely at random [56] showed a non-significant Chi-Square test (p = 0.711), indicating that the data were MCAR. This test was performed for the 35 participants who completed the study. Pattern analysis showed that 0.4% of the data were missing, which seemed negligible.

MI ability is a prerequisite for performing MI effectively, which is why MI abilities were scrutinised in detail at baseline. According to the MIQ-R cut-off point of 12.5 points, 1 out of 35 participants reported low kinaesthetic MI ability, and all participants reported good visual MI ability. There was a negligible to weak and non-significant correlation for some of the MI ability measures. A weak to moderate correlation was observed between the HLJ times for the left (non-dominant) hand and the visual and kinaesthetic subscales of the MIQ-R and also between the HLJ times and MIQ-R subscales and within these measures, all of which were significant at the 0.01 level. In the intervention groups, these relationships were slightly more pronounced, and there was a moderate correlation between the MC measures only in group 2. In addition, in group 2, a strong correlation between the HLJ time and accuracy for the right (dominant) hand was seen (p<0.01). Correlations between the visual and kinaesthetic MIQ-R subscales and of these with the HLJ time for the left hand were also strong (negative) and significant (p<0.01). Moderate relationships shown for the control group were significant (p<0.01) (Fig 2).

Between baseline and post-intervention, there were no statistically significant changes in the MI ability on any of the measures or between groups (p>0.05), but medium effect sizes were observed in visual and kinaesthetic MI abilities (Fig 3). Additionally, medium effect sizes in MC of walking were seen after relaxing MI whereas for the remaining measures of MI capability, effect sizes were negligible to small (S2 Table).

Between baseline and post-intervention, no significant differences were found in any of the metabolic function measures between the groups (p>0.05) (Fig 4). Except medium effects of relaxing MI on BMR, negligible to small effect sizes were observed (S2 Table).

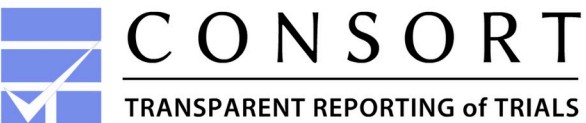

## CONSORT 2010 Flow Diagram

**Enrolment**

Assessed for eligibility (n=362)

Not meeting inclusion criteria – having a chronic disease (n=3)

Meeting inclusion criteria (n=359)

- refused to participate (n=19)
- other reasons (participating in another study) (n=54)
- other reasons (lack of time due to lectures, exams etc. (n= 232)

Eligible before randomisation (n=54)

**Allocation**

Allocated to intervention 1 (n=18)
- received allocated intervention and phone call after week 1 (n=10)
- did not receive allocated intervention (n=8)

Allocated to intervention 2 (n=18)
- received allocated intervention and phone call after week 1 (n=10)
- did not receive allocated intervention (n=8)

Allocated to control group 3 (n=18)
- received phone call after week 1 (n=15)
- did not receive phone call after week 1 (n=3)

**Follow-up**

Lost to follow-up (n=8)
Discontinued intervention (n=8)

Lost to follow-up (n=8)
Discontinued intervention (n=8)

Lost to follow-up (n=3)
Discontinued intervention (n=3)

**Analysis**

Analysed in originally allocated groups (n=10)
Excluded from analysis (n=8)

Analysed in originally allocated groups (n=10)
Excluded from analysis (n=8)

Analysed in originally allocated groups (n=15)
Excluded from analysis (n=3)

**Fig 1. CONSORT flow diagram.**

Between baseline and post-intervention, there were no significant differences in any of the cardiorespiratory function measures between groups (p>0.05) (Fig 5). A majority of effects were negligible to small, but those for $VO_2$ and $FECO_2$ following relaxing MI were medium (S2 Table).

For assessing the acceptability of the intervention, five participants in the activating MI group and 4 in the relaxing MI group took part in the semi-structured interviews. The mean duration of the interviews was 23.9 ± 4.2 minutes. All participants reported to have closed their eyes during the MI. Overall, all participants accepted the intervention and 6 of 9 even regarded it pleasurable and valuable. Seven major themes emerged from the analysis, which are described in detail and together with relevant categories in S2 File. Participants also expressed their opinions on how the use of MI should be for them to continue after the study. Minor themes and their frequencies are shown in Fig 6.

## Discussion

This randomised controlled pilot trial demonstrated the feasibility of a full-scale RCT of MI practice focusing on autonomic function in healthy young people. We hypothesised that young and healthy participants would be able for MI. Study findings showed slightly discrepant results. While the self-report visual and kinaesthetic MI capabilities indicated excellent abilities to imagine movements according to the MIQ-R cut-off, the writing and walking MC tests yielded contradictory results. Median MC scores for the walking test signified perfect temporal congruence in our participants. These results are in agreement with the literature showing that MC is well preserved in young adults as compared to older adults [64]. For the writing task, however, we observed temporal incongruence. Contrastingly, a previous study reported no differences between the MC of walking and writing tasks [41]. A comparison between the two studies showed that both were small, with 35 participants in our study and 16 participants in the previous study. We compared the duration of a single trial of executed and imagined walking and writing tasks; the other study however, employed a repeated trial design,

**Table 1. Demographic characteristics and body composition.**

| Parameter | Activating MI (Gp. 1, N = 10) | Relaxing MI (Gp. 2, N = 10) | Control group (Gp. 3, N = 15) |
|---|---|---|---|
| Gender (females; males)[1] | 9 (90.0); 1 (10.0) | 8 (80.0); 2 (20.0) | 11 (73.3); 4 (26.7) |
| Age (years)[2] | 20.4 (19.5–21.3) | 21.00 (19.7–22.3) | 19.7 (18.9–20.6) |
| Height (m)[2] | 1.63 (1.56–1.70) | 1.66 (1.61–1.70) | 1.65 (1.58–1.72) |
| Weight (kg)[2] | 59.7 (51.4–68.0) | 58.4 (51.6–65.1) | 67.7 (62.0–73.3) |
| Handedness (R; L)[1] | 10 (100.0); 0 (0.0) | 10 (100.0); 0 (0.0) | 14 (93.3); 1 (6.7) |
| **Body composition measures** | | | |
| BMI (kg/m$^2$)[2]* | 22.3 (19.8–24.9) | 21.2 (19.1–23.4) | 24.9 (23.0–26.8) |
| Fat (%)[1] | 22.5 (16.6–28.4) | 22.6 (16.5–28.6) | 27.8 (23.6–32.1) |
| Muscle (%)[1] | 73.3 (68.4–78.2) | 73.1 (68.0-78-3) | 68.5 (64.5–72.6) |
| Water (%)[1] | 55.2 (51.1–59.3) | 57.7 (52.5–62.9) | 52.4 (49.1–55.6) |
| Visceral Fat[1]* | 1.7 (0.7–2.3) | 1.7 (0.5–2.9) | 2.9 (1.9–3.8) |

BMI, Body mass index; Gp., group; F, female; M, male; MI, motor imagery; L, lefthanded; R, righthanded, as assessed by the Edinburgh Handedness Inventory.

[1]Frequency (%) of participants.

[2]Mean (95% confidence interval).

**Table 2. Motor imagery ability and autonomic function at baseline.**

| Parameters | Activating MI (Gp. 1, N = 10) | Relaxing MI (Gp. 2, N = 10) | Control group (Gp. 3, N = 15) |
|---|---|---|---|
| **Motor imagery ability measures** | | | |
| MIQ-R vis[1] | 5.9 (5.5–6.2) | 6.1 (4.7–6.5) | 6.0 (5.0–6.7) |
| MIQ-R kin[1] | 5.6 (4.7–5.7) | 5.9 (5.0–6.5) | 5.7 (5.2–6.0) |
| MC walking (6MWT)[2] | 1.1 (0.8–1.4) | 1.1 (0.8–1.4) | 1.2 (1.0–1.4) |
| MC Writing[2] | 0.7 (0.5–0.8) | 0.6 (0.5–0.8) | 0.8 (0.6–1.0) |
| RL discr time L[2] | 1.9 (1.6–2.2) | 2.0 (1.7–2.3) | 1.9 (1.6–2.2) |
| RL discr time R[2] | 1.8 (1.5–2.2) | 1.9 (1.6–2.2) | 1.8 (1.6–2.1) |
| RL discr acc L[2] | 84.0 (872.6–95.4) | 85.0 (77.3–92.7) | 90.0 (85.6–94.4) |
| RL discr acc R[2] | 83.0 (70.0–96.0) | 85.5 (79.1–91.9) | 93.0 (89.2–96.3) |
| **Autonomic function: cardiorespiratory function and metabolism measures** | | | |
| RMR[2] | 1907.5 (1639.1–2175.9) | 1654.7 (1494.2–1815.1) | 1844.9 (1651.6–2038.3) |
| BMR[2] | 1401.2 (1256.4–1546.0) | 1386.6 (1248.5–1524.7) | 1524.3 (1386.5–1662.1) |
| Fat metabol[2] | 60.2 (48.4–71.9) | 55.6 (37.4–73.9) | 57.2 (43.5–70.9) |
| CHO metabol[2] | 39.8 (28.1–51.5) | 44.3 (26.1–62.6) | 42.8 (29.1–56.5) |
| RQ[2] | 0.8 (0.8–0.9) | 0.8 (0.8–0.9) | 0.8 (0.8–0.9) |
| $VO_2$[2] | 275.4 (237.8–313.0) | 237.4 (214.6–260.2) | 265.8 (237.7–293.9) |
| $VCO_2$[2] | 228.3 (191.5–265.1) | 203.0 (175.6–230.4) | 222.1 (197.2–247.0) |
| VE[2] | 8.9 (7.6–10.3) | 8.5 (7.0–9.9) | 8.1 (7.3–8.9) |
| RF[2] | 17.9 (16.0–19.8) | 16.5 (13.5–19.5) | 16.7 (15.5–18.0) |
| $FEO_2$[2] | 17.2 (17.1–17.4) | 17.5 (17.0–17.9) | 17.0 (16.8–17.2) |
| $FECO_2$[2] | 3.2 (3.0–3.3) | 3.0 (2.7–3.3) | 3.4 (3.2–3.6) |

BMI, body mass index; BMR, basal metabolic rate (kcal); CHO, metabol carbohydrate substrates used for energy metabolism (%); Fat %, body fat percentage; Fat metabol, fat substrates used for energy metabolism (%); $FECO_2$, concentration of carbon dioxide in the exhaled gases (%); $FEO_2$, concentration of oxygen in the exhaled gases (%); MC Walking (6MWT), mental chronometry using a 6-Metre Walk Test; MC Writing, mental chronometry using a writing task; MIQ-R kin, Motor Imagery Questionnaire-Revised, kinaesthetic subscale (median values); MIQ-R vis, Motor Imagery Questionnaire-Revised, visual subscale (median values); Muscle %, body muscle percentage; RF, respiratory frequency; RL discr time L/R, right-left discrimination time for the left/right hand; RL discrim acc L/R, right-left discrimination accuracy for the left/right hand; RMR, resting metabolic rate—caloric expenditure (kcal/day); RQ, respiratory quotient; $VCO_2$, carbon dioxide production (ml/min); VE, minute ventilation (l/min); $VO_2$, oxygen uptake (ml/min); Water %, body water percentage.

[1] Median (25th-75th percentiles).

[2] Mean (95% confidence interval).

with 50 trials (10 actual and 40 imagined) for both the walking and writing tasks. Considering the repeated trial design, a learning effect could have occurred [65], which would explain some of the discrepancies in results. Consistent with the literature, we also suggest that the timing of a cyclical motor task such as walking is easier than that of a complex task like writing [66]. Moreover, walking is a highly automatic motor activity whereas nowadays, writing usually involves typing on a computer keyboard, particularly in young people. As for the HLJ times, our results agree with those from a meta-analysis of 25 studies with found values of 2.08 seconds and 91.9% for the HLJ time and accuracy in healthy controls, respectively [46].

We further hypothesised that there would be moderate to high correlations between the MI ability measures. Results confirmed our study hypotheses partly, with moderate to high and significant correlations between some of the measures. Another study assessed MI abilities in 12 swimmers using electrodermal (skin resistance) responses, subjective ratings (Visual Imagery of Movement Questionnaire-2), and an MC test (swimming turn sequence) [67]. Results showed significant moderate correlations between MI vividness and autonomic responses during MI, while all other correlations were non-significant [67]. Our findings are in line with those findings.

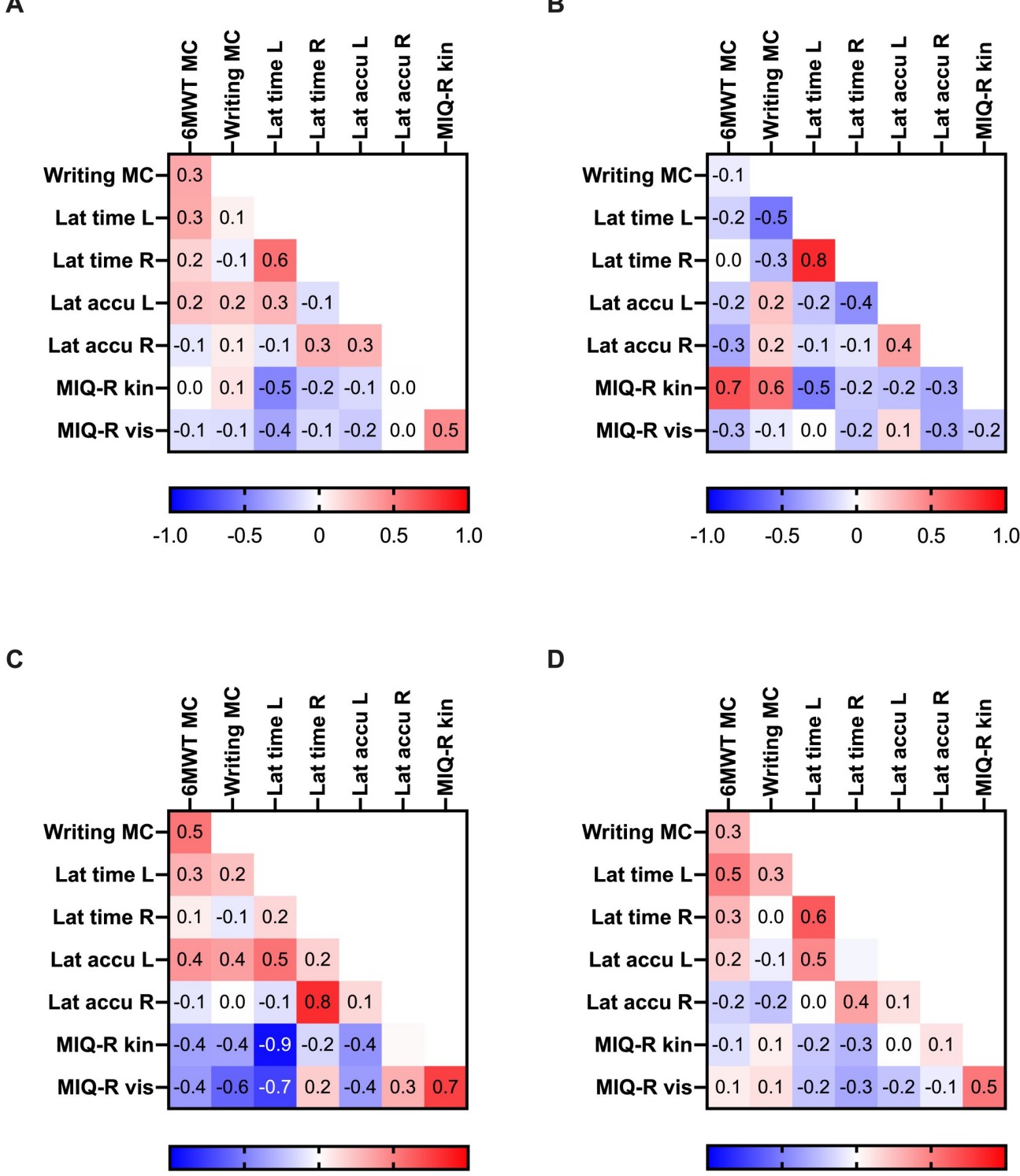

**Fig 2. Correlations between the motor imagery ability assessments at baseline.** The heatmap presents Spearman's Rank correlation coefficients (ranging from -1 to +1) of MI ability measures, where darker blue and red fields signify stronger correlations. (**A**) All participants. (**B**) Activating MI group. (**C**) Relaxing MI group. (**D**) Control group.

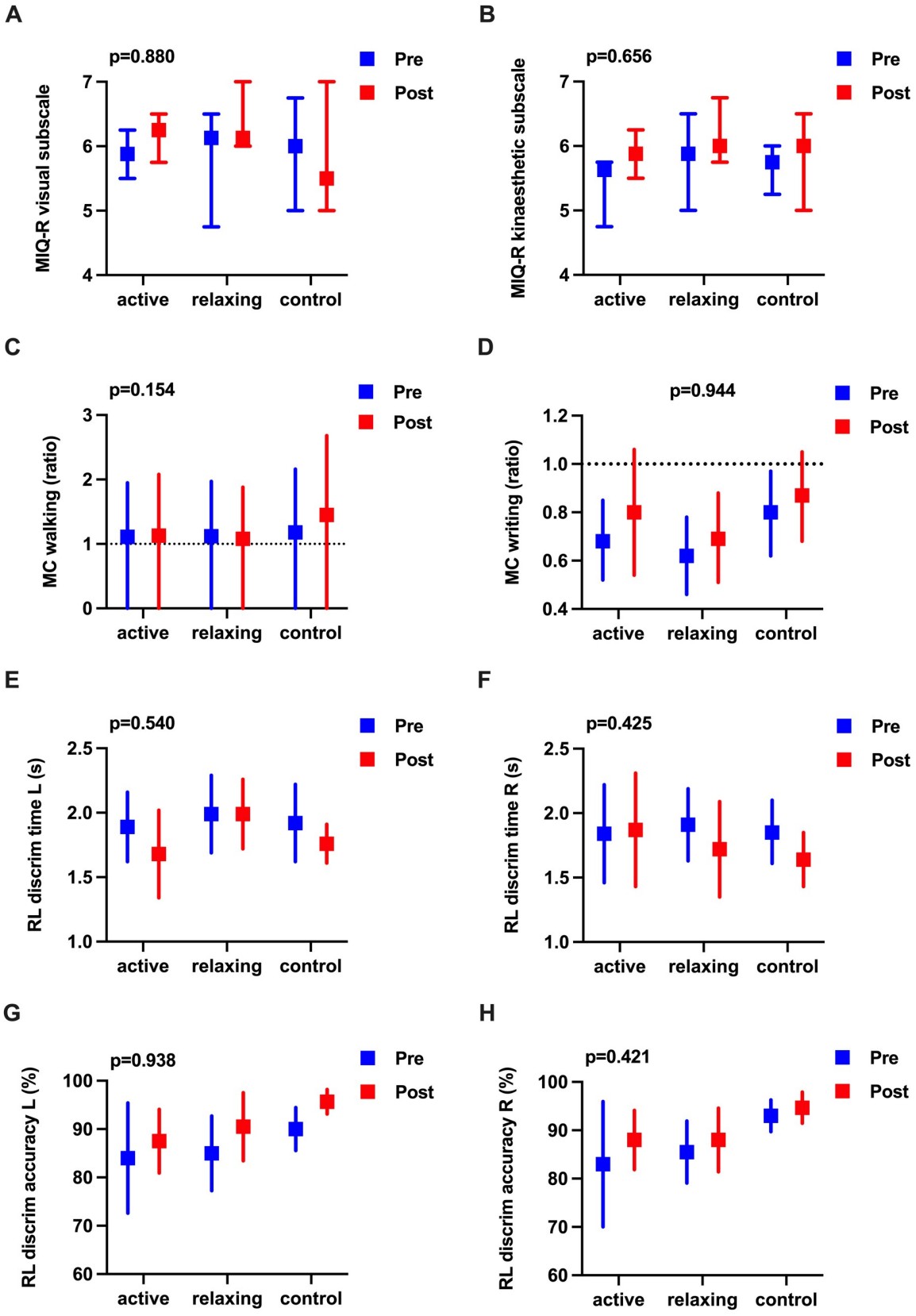

**Fig 3. Effects of activating and relaxing motor imagery ability on motor imagery ability.** (**A**) Visual MI ability. (**B**) Kinaesthetic MI ability. (**A and B**) Medians at baseline (pre) and post-intervention (post) are represented by squares and interquartile ranges by error bars. Units of measurement are indicated on the y-axis. Motor Imagery Questionnaire-Revised (MIQ-R) data were analysed using Kruskal-Wallis test, followed by Dunn's correction for multiple comparisons, with overall p-values and partial eta squared effect sizes shown on top of the figures. (**C to H**) Means at baseline (pre) and post-intervention (post) are represented by squares and lower and upper 95% confidence intervals by error bars. Units of measurement are indicated on the y-axis. Mental chronometry (MC) and hand laterality judgements (HLJ) were analysed using a two-factor mixed analysis of variance ANOVA, followed by a Bonferroni correction for multiple comparisons, with overall p-values shown on top of the figures. (**C**) MC between imagined and real walking using a 6-Metre Walk Test, where the dotted line indicates identical durations of real and imagined walking. (**D**) Mental chronometry using a writing task, where the dotted line indicates identical durations of real and imagined writing. (**E to F**) HLJ. (**E**) Right-left discrimination (RLD) time (seconds) for the left hand. (**F**) RLD time (seconds) for the right hand. (**G**) RLD accuracy (percentage of correct responses) for the left hand. (**H**) RLD accuracy (percentage of correct responses) for the right hand.

Results from the qualitative interview analyses showed that participants accepted or liked the interventions in both study groups. This is crucial indeed because one cannot expect people maintaining any practice of real or imagined exercises which they do not appreciate.

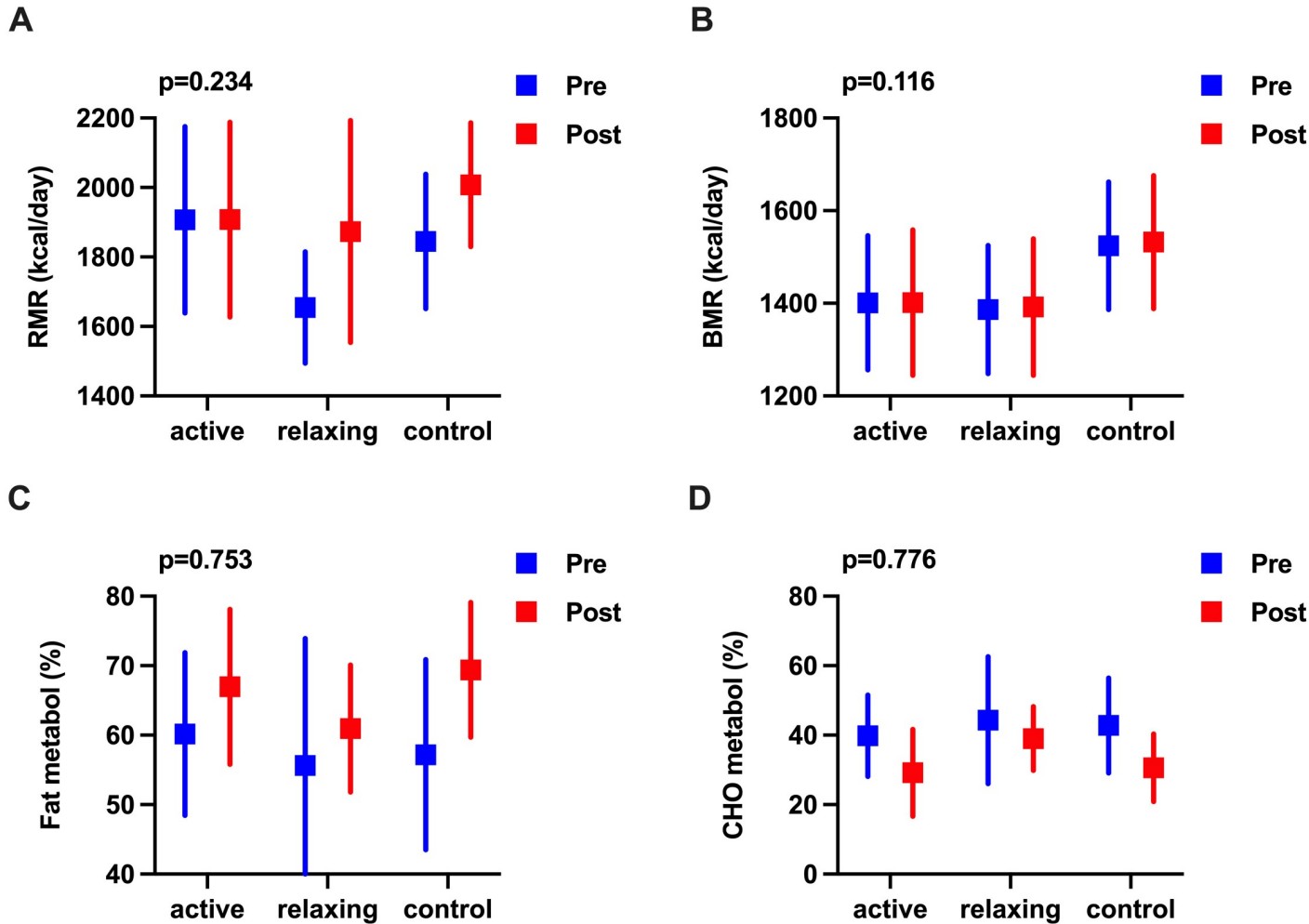

**Fig 4. Effects of activating and relaxing motor imagery ability on metabolic function. (A to D)** Means at baseline (pre) and post-intervention (post) are represented by squares and lower and upper 95% confidence intervals by error bars. Units of measurement are indicated on the y-axis. Data were analysed using a two-factor mixed analysis of variance ANOVA, followed by a Bonferroni correction for multiple comparisons, with overall p-values shown on top of the figures. (**A**) RMR resting metabolic rate—caloric expenditure (**B**) BMR basal metabolic rate. (**C**) Fat metabol fat substrates used for energy metabolism. (**D**) CHO metabol carbohydrate substrates used for energy metabolism.

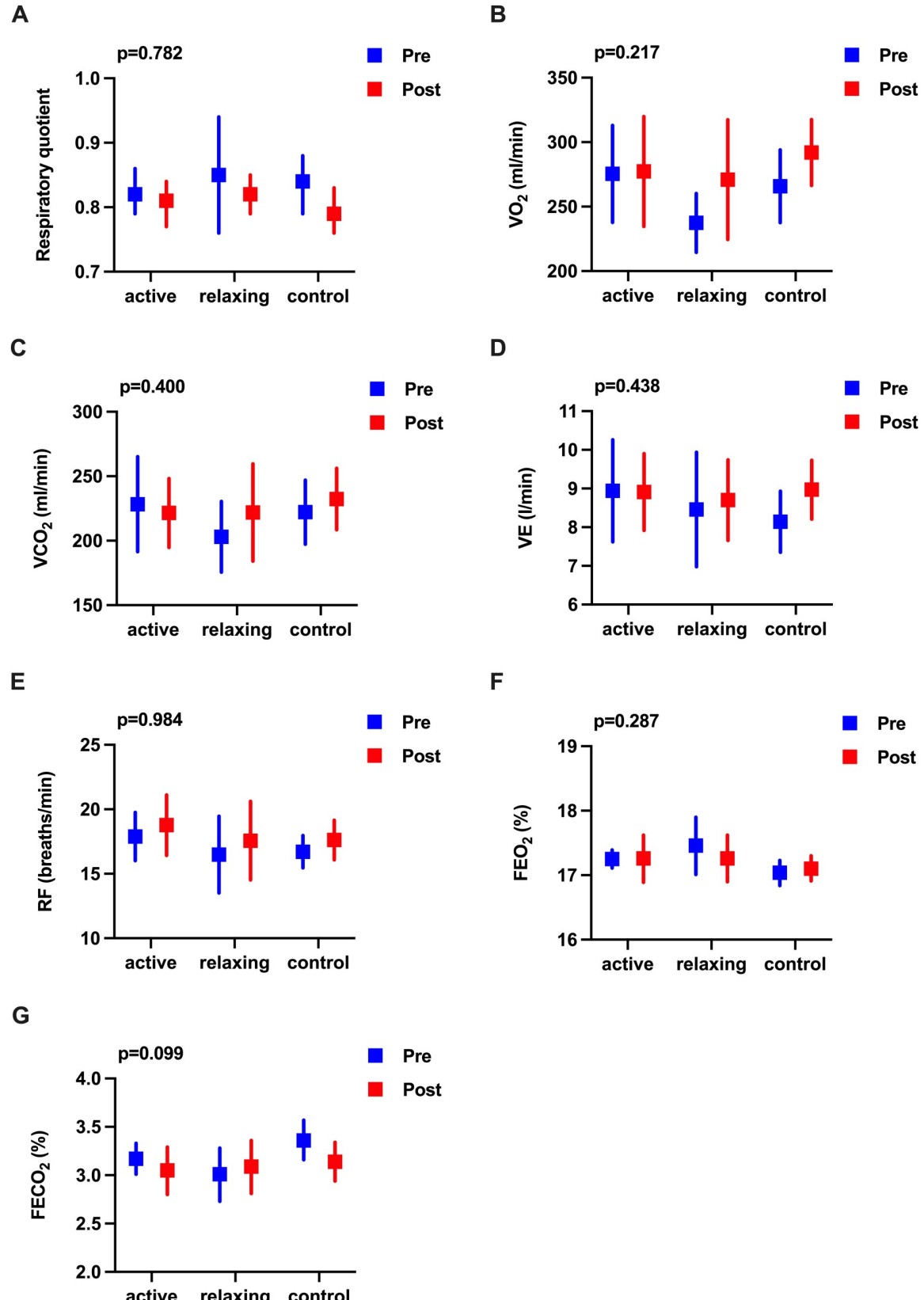

**Fig 5. Effects of activating and relaxing motor imagery ability on cardiorespiratory function. (A to G)** Means at baseline (pre) and post-intervention (post) are represented by squares and lower and upper 95% confidence intervals by error bars. Units of measurement are indicated on the y-axis. Data were analysed using a two-factor mixed analysis of variance ANOVA, followed by a Bonferroni correction for multiple comparisons, with overall p-values shown on top of the figures. **(A)** RQ respiratory quotient. **(B)** $VO_2$ oxygen uptake. **(C)** $VCO_2$ carbon dioxide production. **(D)** VE minute ventilation. **(E)** RF respiratory frequency. **(F)** $FEO_2$ concentration of oxygen in the exhaled gases. **(G)** $FECO_2$ concentration of carbon dioxide in the exhaled gases.

Findings also indicated that the participants did experience MI; however, using a kinaesthetic or visual or mixed-mode and an internal or external perspective. Some of the participants explained that it was more natural for them to visually imagine themselves moving. Based on that, we speculate that the use of an alternative mode or perspective enabled them for MI or enhanced their MI capability. This was supported by their descriptions of their body perceptions, for example to having felt the muscles quite intensely. Further, their narratives on perceived barriers to MI such as fatigue and loss of concentration were in line with the literature [68, 69]. Facilitators were reported such as a quiet environment and being alone, which would naturally enhance concentration. A learning effect was also mentioned as some participants described initial difficulties with imagining the tasks which resolved with practice. Interestingly, participants in the activating MI group expressed some problems with MI of unfamiliar

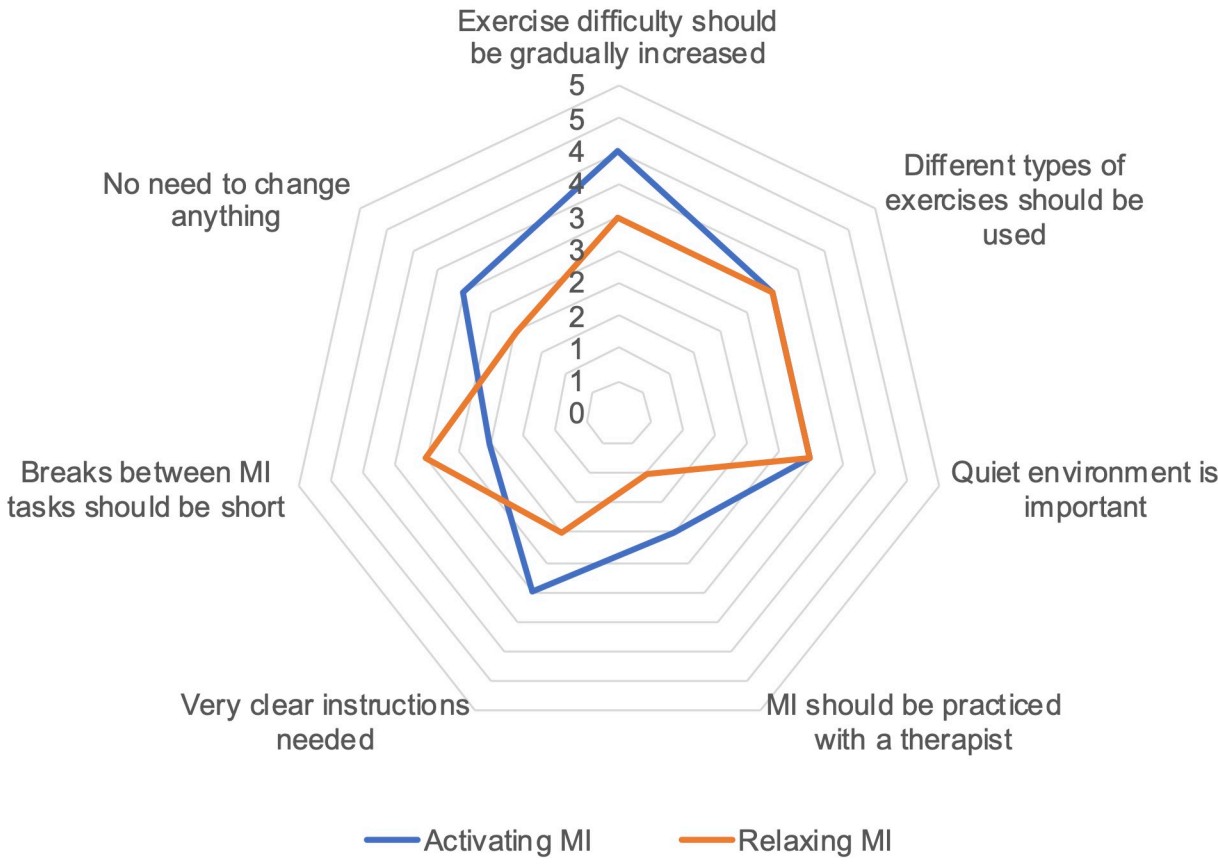

**Fig 6. Motor imagery recommendations from study participants.** Numbers on the radar graph represent the numbers of participants in the two intervention groups who expressed the respective opinions.

movements, particularly if they could not perform these motor tasks in real e.g., due to a lack of muscle strength. A potential lack in fitness had not been considered during the study development, since all participants were young and healthy physiotherapy students. In addition, we observed medium effects in mental chronometry of walking after relaxing MI only. Although walking was none of the trained tasks, the relaxed MI training of various movements with a focus on body perception could have contributed to this change. However, the findings were in agreement with the existing literature in people with neurological disorders such as stroke [69]. Additionally, a study in young dancers has found that with complex and unfamiliar movements e.g., complex jumps, primarily visual MI was used whereas kinaesthetic MI and real movement durations were significantly longer [70].

However, one could question whether a temporal congruence between both the visual and kinaesthetic MI modes and executed motor actions was required in all cases. Considering the theoretical tenets of MI, with the notion of functional equivalence [8], MC may indeed be a prerequisite. It is established that, prior to the execution of a motor action, the brain has already generated a motor representation, which is thought to include all movement phases from the movement plan until the intended result [71]. It seems logical that for a completely unknown motor action, motor representations are not yet available [72]. So, a reasonable level of familiarity and expertise gained by the training of a movement appears essential for the functional equivalence between physical and imagined movements. Evidence suggests that it may not be possible to imagine a motor task appropriately if one cannot execute the task physically [72]. This proposition has been supported by a functional magnetic resonance imaging study from Olsson et al. (2008) conducted in expert and novice high jumpers [73]. Results showed that MI of the complex high jumps led to activations of motor areas only in expert jumpers whereas in novices, recruitment of the visual and parietal areas was seen.

In our study, activating and relaxing MI training did not lead to statistically significant preliminary effects on autonomic function. This was opposed to our hypothesis that at least trends for a change in autonomic function would be seen post-intervention, with contrasting effects after activating and relaxing MI and as compared to the control group. Our hypothesis was based on the functional equivalence between real motor actions and imagined motor tasks. Functional equivalence provided, a high-intensity and relaxing MI training would yield similar but weaker effects than a real training of the same motor activities [74]. Concerning actual training, numerous studies have shown that both high-intensity training and relaxation practice [75] induce significant changes in autonomic function [76]. It is acknowledged that the present study was small due to the COVID-19 pandemic, as mentioned above. Therefore, there may have been a lack of power to detect significant differences. This suggestion is strengthened by medium effect sizes found for differences in $VO_2$ and $FECO_2$ after relaxing MI versus activating and controls. Nonetheless, other factors may be responsible for the absence of relevant changes post-intervention, which are yet to be explored.

A study compared the effects of MI alone with MI plus action observation against a control group on autonomic function in 45 healthy people and showed differences in the heart and respiratory rates and some of the measures of electrodermal activities [77]. Although different autonomic function measures were used in our study, one would expect similar results. However, that study investigated changes after a single intervention session. Conceivably, there may have been a habituation effect in our study, associated with the MI practice of five times per week and over two weeks. The autonomic nervous system is sensitive to novel stimuli and hence produces stronger responses such as heart rate increases alongside physical reactions [78]. If the same stimuli were presented repeatedly, like MI tasks in our study, elicited autonomic responses would be weaker or even absent because of habituation [79]. Moreover, in contrast to a previous study [80], we did not assess autonomic functions during the MI.

Autonomic responses are often short-lived, a notion supported by another MI study in elite athletes, where steep increments in heart rates were seen when the MI commenced [80]. However, long-term changes in the resting heart rate after strength and endurance training, yoga, and tai chi have been reported by various studies [81], and hence, we regarded it reasonable to anticipate changes in autonomic responses after the intervention in our study.

There are some limitations to our study. Firstly, in this study, convenience sampling was used because it is straightforward, low-cost and efficient. For a larger study in a normal sample, considering the lack of generalisability of convenience samples, it seems useful however, taking into account a higher dropout rate of 30%. Secondly, although the number of participants enrolled was based on a sample size calculation including a 20% attrition rate, due to the COVID-19 pandemic-related shut-down, a substantial attrition rate of 35% was observed. Thus, the study remained with a small sample size. We accounted for that examining the type and patterns of missing data. Thirdly, no data were collected during the MI practice. This would have enabled us to compare our results to those from other studies directly. However, we expected to see some trends for changes in autonomic functions after the two-week intervention beyond the MI sessions. In addition, as our study intervention was home-based and involved 10 sessions, such measurements would have been impracticable. Finally, the intervention period was two weeks due to the exploratory nature of this study. Using a longer intervention duration, the chances for long-term effects may have been greater. The intervention was probably too long for short-term responses in the autonomic function to be elicited and too short for long-term effects to occur.

Further studies are required to figure out underlying mechanisms of activating and relaxing MI, particularly concerning the functional equivalence theory, and potentially including functional imaging measures. Future studies should also investigate the effects of MI over a longer period. These studies may assess autonomic function at several time points and simultaneously with activating (high intensity) and relaxing MI. Additional measures of autonomic functions could be useful, such as electrodermal activity.

## Conclusions

Both activating and relaxing MI are acceptable interventions and a larger RCT is feasible. Healthy young people showed excellent MI abilities where the different MI ability measures corresponded with each other. After a two-week intervention consisting of activating or relaxing MI, no changes in the autonomic function were observed compared to non-intervention controls. These findings could be related to a habituation effect after the prolonged MI practice. The absence of measures performed during MI sessions could contribute to the outcomes. Adequately powered studies are needed to unravel the underlying mechanisms of activating and relaxing MI concerning the functional equivalence theory.

## Supporting information

**S1 Checklist. CONSORT checklist.**
(DOCX)

**S2 Checklist. COREQ checklist.**
(DOCX)

**S1 File. PETTLEP principles of the motor imagery interventions.**
(PDF)

**S2 File. Qualitative methods and results.**
(PDF)

**S1 Study protocol. Study protocol English.**
(PDF)

**S1 Table. Questions asked in the semi-structured interviews.**
(PDF)

**S2 Table. Effect sizes of activating and relaxing motor imagery related to all outcomes.**
(PDF)

## Acknowledgments

We would like to warmly thank all the participants in this study, Professor Markus Reindl for his help with the graphics, Marilena Politaki for helping with the videos and Dr Buse Ozcan Kahraman for her support during the preparation and vocalisation of Turkish scripts.

## Author Contributions

**Conceptualization:** Turhan Kahraman, Barbara Seebacher.

**Formal analysis:** Barbara Seebacher.

**Funding acquisition:** Turhan Kahraman.

**Investigation:** Turhan Kahraman, Tayfun Isik, Sukriye Cansu Gultekin.

**Methodology:** Turhan Kahraman, Derya Ozer Kaya, Barbara Seebacher.

**Project administration:** Turhan Kahraman, Tayfun Isik, Sukriye Cansu Gultekin.

**Resources:** Turhan Kahraman, Derya Ozer Kaya, Barbara Seebacher.

**Software:** Barbara Seebacher.

**Supervision:** Turhan Kahraman.

**Validation:** Barbara Seebacher.

**Writing – original draft:** Turhan Kahraman, Barbara Seebacher.

**Writing – review & editing:** Turhan Kahraman, Derya Ozer Kaya, Tayfun Isik, Sukriye Cansu Gultekin, Barbara Seebacher.

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
