## [Decision Letter · Decision Letter 0]

5 May 2021

PONE-D-21-06610

Feasibility of motor imagery and effects of activating and relaxing practice on autonomic functions in healthy young adults: a randomised, controlled, assessor-blinded, pilot trial

PLOS ONE

Dear Dr. Seebacher,

Thank you for submitting your manuscript to PLOS ONE. After careful consideration, we feel that it has merit but does not fully meet PLOS ONE’s publication criteria as it currently stands. Therefore, we invite you to submit a revised version of the manuscript that addresses the points raised during the review process.

We look forward to receiving your revised manuscript.

Kind regards,

Walid Kamal Abdelbasset, Ph.D.

Academic Editor

PLOS ONE

Journal Requirements:

Reviewers' comments:

Reviewer's Responses to Questions

**Comments to the Author**

1. Is the manuscript technically sound, and do the data support the conclusions?

Reviewer #1: Yes

Reviewer #2: Yes

Reviewer #3: Partly

2. Has the statistical analysis been performed appropriately and rigorously? 

Reviewer #1: No

Reviewer #2: Yes

Reviewer #3: No

3. Have the authors made all data underlying the findings in their manuscript fully available?

Reviewer #1: Yes

Reviewer #2: Yes

Reviewer #3: No

4. Is the manuscript presented in an intelligible fashion and written in standard English?

Reviewer #1: Yes

Reviewer #2: Yes

Reviewer #3: Yes

5. Review Comments to the Author

Reviewer #1: Generally well conducted study. I will focus on methods and reporting. I only have a few points to make.

Major

1) clearly state the outcome in a relevant section (or edit the MI section to include it at the start) and do the same for the abstract

2) in the methods section I couldn't find any information on analysing the outcome(s) of interest, in a repeated measures design (i.e. logistic regression, clustering for patient, ANOVA etc)

Minor

1) The power calculations are not replicable at the minute. please include all information relevant to the power calculations so they are replicable.

2) Convenience sampling is obviously problematic for generalisability, so the authors should consider potentially a higher drop off if they recruit a normal sample.

3) define MCAR

Reviewer #2: -

Reviewer #3: The manuscript entitled ‘Feasibility of motor imagery and effects of activating and relaxing practice on autonomic functions in healthy young adults: a randomised, controlled, assessor blinded, pilot trial’ with the aims to investigate the feasibility of MI training focusing on the autonomic function in healthy young people and to evaluate participants’ MI abilities and compare the preliminary effects of activating and relaxing MI on autonomic function and against a control group.

The manuscript can be further improved based on the following comments.

Abstract

Page 2 Line 33, the sentence ‘feasibility of a full-scale randomised controlled trial using predefined feasibility’ requires improvement.

Materials and Methods-Participants

Page 6 Line 113, the sentence not clear and requires revision.

Page 6 Line 114, 1 or 2-tailed test to be stated.

Motor imagery ability assessment

Page 11 Line 234, 1-((actual movement duration/MI duration)/actual movement duration), the bracket to be revised.

Page 12 Line 243, for 91.9 (95%CI: 91.3, 92.5)%, % to be placed after the figures.

Statistical analysis

Page 13 Line 285, the statement ‘Kruskal–Wallis test was used on the difference measures (new variables) between post-intervention and baseline values’ not clear and requires revision.

Results

Page 14 Table 1, based on CONSORT statement, all statistical tests for baseline comparison between the groups to be avoided.

Frequency (%) to be displayed in table for chi-square test. 95% CI to be omitted,

Page 15 Table 2, RF to be denoted with superscript ‘1’ . If all data are presented as median (IQR), symbol superscript ‘1’ can be omitted.

Page 17 Line 318-319 & 326-327, for FeO2, FECO2, VO2 and VCO2, 2 to be subscripted.

Page 17 Line 340, random[56], word random and reference to be spaced out.

The decimal points for percentage figures to be standardized. Likewise with correlation coefficient values.

For Figure 1, eligible numbers before the randomization and assessment period to be stated. Intent to treat to be stated for both Figure 1 and in tables footnote where applicable.

For Figure 2, it would be good to provide a table to display the figures at baseline and post intervention, the difference of pre and post intervention, effect size, 95% CI and p value. Likewise for Figure 3.

For Figure 3, A, B, C, D to be denoted.

The focus of the analysis to be more emphasized on the pre and post intervention of each group than at baseline and at post intervention respectively between the groups.

Some references in the manuscript and supplementary documents did not conform to PLoS One format

6. PLOS authors have the option to publish the peer review history of their article (what does this mean?). If published, this will include your full peer review and any attached files.

Reviewer #1: No

Reviewer #2: No

Reviewer #3: No

---

## [Author Response · Author response to Decision Letter 0]

20 May 2021

5. Review Comments to the Author

Reviewer #1: Generally well conducted study. I will focus on methods and reporting. I only have a few points to make.

Dear Reviever #1, Thank you for your suggestions, which helped us to improve the quality of our manuscript. Please see our point-by-point responses as follows.

Major

1) clearly state the outcome in a relevant section (or edit the MI section to include it at the start) and do the same for the abstract

We have stated the primary and secondary outcomes in a relevant section. Please see the tracked changes in the manuscript.

2) in the methods section I couldn't find any information on analysing the outcome(s) of interest, in a repeated measures design (i.e. logistic regression, clustering for patient, ANOVA etc)

Thank you for this comment. We have now included information on a two-factor mixed analysis of variance (ANOVA) in the Statistical analysis and results sections, including Figures 3-5 (new) and S2 Table showing effect sizes. 

Minor

1) The power calculations are not replicable at the minute. please include all information relevant to the power calculations so they are replicable.

We did accordingly, so that the sample size calculations now read as follows: Twelve participants per group are considered a minimum sample size [1], and 10% of the total sample size of a full-scale study is suggested for pilot studies [2]. Based on a power of 80%, a two-tailed alpha of 0.05, mean 1 of -2.13, mean 2 of 0.20 and pooled standard deviation of 4.67 from the group differences in respiratory rates of an MI training study that used physiological measures [3], a sample size of 189 participants for three groups resulted (http://powerandsamplesize.com/). Considering these recommendations, the sample size was determined as 15 participants per group (by addition of 20% attrition, a total of 54 participants).

2) Convenience sampling is obviously problematic for generalisability, so the authors should consider potentially a higher drop off if they recruit a normal sample.

Thank you for this suggestion. We have included the following limitation in our Discussion section: Firstly, in this study, convenience sampling was used because it is straightforward, low-cost and efficient. For a larger study in a normal sample, considering the lack of generalisability of convenience samples, it seems useful however, taking into account a higher dropout rate of 30%.

3) define MCAR

We did accordingly: Little’s test was performed to evaluate whether the data were missing completely at random (MCAR).

Reviewer #2: -

Reviewer #3: The manuscript entitled ‘Feasibility of motor imagery and effects of activating and relaxing practice on autonomic functions in healthy young adults: a randomised, controlled, assessor blinded, pilot trial’ with the aims to investigate the feasibility of MI training focusing on the autonomic function in healthy young people and to evaluate participants’ MI abilities and compare the preliminary effects of activating and relaxing MI on autonomic function and against a control group.

The manuscript can be further improved based on the following comments.

Dear Reviewer #3, Many thanks for your suggestions, which help us to improve the quality of our manuscript. Please find our point-by-point responses as follows.

Abstract

Page 2 Line 33, the sentence ‘feasibility of a full-scale randomised controlled trial using predefined feasibility’ requires improvement.

We have revised the sentence so that it now reads: The primary outcome was the feasibility of a full-scale randomised controlled trial using predefined criteria.

Materials and Methods-Participants

Page 6 Line 113, the sentence not clear and requires revision.

We revised the description of the sample size calculation so that it now reads: Twelve participants per group are considered a minimum sample size [1], and 10% of the total sample size of a full-scale study is suggested for pilot studies [2]. Based on a power of 80%, a two-tailed alpha of 0.05, mean 1 of -2.13, mean 2 of 0.20 and pooled standard deviation of 4.67 from the group differences in respiratory rates of an MI training study that used physiological measures [3], a sample size of 189 participants for three groups resulted (http://powerandsamplesize.com/). Considering these recommendations, the sample size was determined as 15 participants per group (by addition of 20% attrition, a total of 54 participants).

Page 6 Line 114, 1 or 2-tailed test to be stated.

We did accordingly.

Motor imagery ability assessment

Page 11 Line 234, 1-((actual movement duration/MI duration)/actual movement duration), the bracket to be revised.

We checked the formula with the original formula from Collet et al., 2011 [4] and found that the first fraction should be a difference. In our calculation, this was correct. In the manuscript, we replaced / by -. After this revision, the brackets should be correct.

Page 12 Line 243, for 91.9 (95%CI: 91.3, 92.5)%, % to be placed after the figures.

We did accordingly.

Statistical analysis

Page 13 Line 285, the statement ‘Kruskal–Wallis test was used on the difference measures (new variables) between post-intervention and baseline values’ not clear and requires revision.

We revised this sentence and stated that Kruskal-Wallis test was used across all groups (Gps 1-3) and time points (baseline, post-intervention), followed by Dunn’s multiple comparisons test.

Results

Page 14 Table 1, based on CONSORT statement, all statistical tests for baseline comparison between the groups to be avoided.

We have eliminated the p-values column in Table 1 and also removed the description within the text. Likewise, we have also deleted p-values in Table 2 showing the baseline comparison between groups for motor imagery ability and autonomic function. In the Statistical analysis section, we referred to the CONSORT statement and deleted the description of statistical tests for baseline comparison between groups.

Frequency (%) to be displayed in table for chi-square test. 95% CI to be omitted,

Thank you for this suggestion as unfortunately, we had used a wrong superscript. For variables of gender and handedness, we have now used frequency (%) and for continuous variables, we have used the mean (95% confidence interval). According to comments from Reviewer #1, we have reanalysed all continuous data using a 2-Way Mixed Design ANOVA (partly after reciprocal transformation); for reasons of consistency, we have shown all continuous data as mean (95% CI).

Page 15 Table 2, RF to be denoted with superscript ‘1’ . If all data are presented as median (IQR), symbol superscript ‘1’ can be omitted.

In line with what we stated for Table 1, we have now presented continuous variables in Table 2 as mean (95% CI). As ordinal variables are still presented as median (25th-75th percentile), we have kept the superscripts 1 and 2 respectively.

Page 17 Line 318-319 & 326-327, for FeO2, FECO2, VO2 and VCO2, 2 to be subscripted.

We did accordingly. Please also see our explanation as to the presentation of variables in Tables 1 and 2 above.

Page 17 Line 340, random[56], word random and reference to be spaced out.

We did accordingly.

The decimal points for percentage figures to be standardized. Likewise with correlation coefficient values.

We have now used one decimal place for all tables, percentage figures and correlation coefficient values (the latter had already been standardised).

For Figure 1, eligible numbers before the randomization and assessment period to be stated. Intent to treat to be stated for both Figure 1 and in tables footnote where applicable.

We have revised Figure 1, so that it now clearly states the eligible numbers before the randomisation and assessment period. Concerning intention-to-treat, we included a description according to the CONSORT statement i.e., having analysed participants in originally allocated groups. For tables, this was not applicable because we decided to report effects of activating and relaxing motor imagery in figures (please see our explanation on that with our response to your next comment).

For Figure 2, it would be good to provide a table to display the figures at baseline and post intervention, the difference of pre and post intervention, effect size, 95% CI and p value. Likewise for Figure 3.

We have reanalysed our data, also according to the comments from Reviewer #1. Thus, the motor imagery data at baseline are presented in Figure 2, as before. In addition, motor imagery ability differences between baseline (pre) and post-intervention (post) are shown in Figure 3 (new). Differences in metabolic function between pre and post and between groups (group by time interaction) are presented in Figure 4 (new). Differences in cardiorespiratory function are shown in Figure 5 (new). All figures include p-values and the 95% CI or IQR (for MIQ-R data). We tried to present all outcome data in a table, but the table was huge (5 pages). This is why we decided to use both the figures and include an S2 Table showing all effect sizes, according to your suggestion.

For Figure 3, A, B, C, D to be denoted.

We did accordingly. 

The focus of the analysis to be more emphasized on the pre and post intervention of each group than at baseline and at post intervention respectively between the groups.

We did accordingly. Please see Figures 3-5 and related descriptions in the statistical analyses, results and discussion sections.

Some references in the manuscript and supplementary documents did not conform to PLoS One format

We updated the references in the manuscript and S1 and S2 files according to the PLoS One format.

6. PLOS authors have the option to publish the peer review history of their article (what does this mean?). If published, this will include your full peer review and any attached files.

Do you want your identity to be public for this peer review? For information about this choice, including consent withdrawal, please see our Privacy Policy.

Reviewer #1: No

Reviewer #2: No

Reviewer #3: No

1. Julious SA. Sample size of 12 per group rule of thumb for a pilot study. Pharmaceutical statistics. 2005;4:287-91. doi: 10.1002/pst.185.

2. Treece EW, Treece JW. Elements of research in nursing. 3rd ed. St. Louis, MO: Mosby; 1982.

3. Wang Y, Morgan WP. The effect of imagery perspectives on the psychophysiological responses to imagined exercise. Behavioural brain research. 1992;52(2):167-74. PubMed PMID: 1294196.

4. Collet C, Guillot A, Lebon F, MacIntyre T, Moran A. Measuring motor imagery using psychometric, behavioral, and psychophysiological tools. Exercise and sport sciences reviews. 2011;39(2):85-92. doi: 10.1097/JES.0b013e31820ac5e0. PubMed PMID: 21206282.

---

## [Decision Letter · Decision Letter 1]

2 Jun 2021

PONE-D-21-06610R1

Feasibility of motor imagery and effects of activating and relaxing practice on autonomic functions in healthy young adults: a randomised, controlled, assessor-blinded, pilot trial

PLOS ONE

Dear Dr. Seebacher,

Thank you for submitting your manuscript to PLOS ONE. After careful consideration, we feel that it has merit but does not fully meet PLOS ONE’s publication criteria as it currently stands. Therefore, we invite you to submit a revised version of the manuscript that addresses the points raised during the review process.

We look forward to receiving your revised manuscript.

Kind regards,

Walid Kamal Abdelbasset, Ph.D.

Academic Editor

PLOS ONE

Journal Requirements:

Reviewers' comments:

Reviewer's Responses to Questions

**Comments to the Author**

1. If the authors have adequately addressed your comments raised in a previous round of review and you feel that this manuscript is now acceptable for publication, you may indicate that here to bypass the “Comments to the Author” section, enter your conflict of interest statement in the “Confidential to Editor” section, and submit your "Accept" recommendation.

Reviewer #1: All comments have been addressed

Reviewer #2: All comments have been addressed

Reviewer #3: (No Response)

2. Is the manuscript technically sound, and do the data support the conclusions?

Reviewer #1: Yes

Reviewer #2: Yes

Reviewer #3: Partly

3. Has the statistical analysis been performed appropriately and rigorously? 

Reviewer #1: Yes

Reviewer #2: Yes

Reviewer #3: (No Response)

4. Have the authors made all data underlying the findings in their manuscript fully available?

Reviewer #1: Yes

Reviewer #2: Yes

Reviewer #3: Yes

5. Is the manuscript presented in an intelligible fashion and written in standard English?

Reviewer #1: Yes

Reviewer #2: Yes

Reviewer #3: Yes

6. Review Comments to the Author

Reviewer #1: I am happy with the changes the authors have made and their responses to the points I previously raised.

Reviewer #2: thanks alot for your response

Reviewer #3: Table 2 Fat2 metabol2: typo 60.2 48.4-71.9)

Line 257, Line 321-322, Line Line 330-331, Table 2 footnote: for FeO2, FeCO2, VCO2, VO2, 2 to be subscripted.

7. PLOS authors have the option to publish the peer review history of their article (what does this mean?). If published, this will include your full peer review and any attached files.

Reviewer #1: No

Reviewer #2: No

Reviewer #3: No

---

## [Author Response · Author response to Decision Letter 1]

3 Jun 2021

PONE-D-21-06610R1

Feasibility of motor imagery and effects of activating and relaxing practice on autonomic functions in healthy young adults: a randomised, controlled, assessor-blinded, pilot trial

PLOS ONE

Reviewers' comments:

Reviewer's Responses to Questions

Comments to the Author

1. If the authors have adequately addressed your comments raised in a previous round of review and you feel that this manuscript is now acceptable for publication, you may indicate that here to bypass the “Comments to the Author” section, enter your conflict of interest statement in the “Confidential to Editor” section, and submit your "Accept" recommendation.

Reviewer #1: All comments have been addressed

Reviewer #2: All comments have been addressed

Reviewer #3: (No Response)________________________________________

2. Is the manuscript technically sound, and do the data support the conclusions?

Reviewer #1: Yes

Reviewer #2: Yes

Reviewer #3: Partly

3. Has the statistical analysis been performed appropriately and rigorously? 

Reviewer #1: Yes

Reviewer #2: Yes

Reviewer #3: (No Response)

4. Have the authors made all data underlying the findings in their manuscript fully available?

Reviewer #1: Yes

Reviewer #2: Yes

Reviewer #3: Yes

5. Is the manuscript presented in an intelligible fashion and written in standard English?

Reviewer #1: Yes

Reviewer #2: Yes

Reviewer #3: Yes

6. Review Comments to the Author

Reviewer #1: I am happy with the changes the authors have made and their responses to the points I previously raised.

Reviewer #2: thanks alot for your response

Reviewer #3: Table 2 Fat2 metabol2: typo 60.2 48.4-71.9)

Line 257, Line 321-322, Line Line 330-331, Table 2 footnote: for FeO2, FeCO2, VCO2, VO2, 2 to be subscripted.

Authors’ response to the Reviewers

Dear Reviewer #1,

Thank you for accepting our changes to the manuscript and responses.

Dear Reviewer #2,

Thank you for accepting our changes to the manuscript and responses.

Dear reviewer #3,

Thank you again for your comments on our manuscript and response. Please find our point-by-point response as follows. 

Table 2 Fat2 metabol2: typo 60.2 48.4-71.9)

We have added the bracket so that it now reads 60.2 (48.4-71.9)

Line 257, Line 321-322, Line Line 330-331, Table 2 footnote: for FeO2, FeCO2, VCO2, VO2, 2 to be subscripted.

We did accordingly.

---

## [Decision Letter · Decision Letter 2]

1 Jul 2021

Feasibility of motor imagery and effects of activating and relaxing practice on autonomic functions in healthy young adults: a randomised, controlled, assessor-blinded, pilot trial

PONE-D-21-06610R2

Dear Dr. Seebacher,

We’re pleased to inform you that your manuscript has been judged scientifically suitable for publication and will be formally accepted for publication once it meets all outstanding technical requirements.

Kind regards,

Walid Kamal Abdelbasset, Ph.D.

Academic Editor

PLOS ONE

Additional Editor Comments (optional):

Reviewers' comments:

Reviewer's Responses to Questions

**Comments to the Author**

1. If the authors have adequately addressed your comments raised in a previous round of review and you feel that this manuscript is now acceptable for publication, you may indicate that here to bypass the “Comments to the Author” section, enter your conflict of interest statement in the “Confidential to Editor” section, and submit your "Accept" recommendation.

Reviewer #2: All comments have been addressed

Reviewer #3: All comments have been addressed

2. Is the manuscript technically sound, and do the data support the conclusions?

Reviewer #2: Yes

Reviewer #3: (No Response)

3. Has the statistical analysis been performed appropriately and rigorously? 

Reviewer #2: Yes

Reviewer #3: (No Response)

4. Have the authors made all data underlying the findings in their manuscript fully available?

Reviewer #2: Yes

Reviewer #3: (No Response)

5. Is the manuscript presented in an intelligible fashion and written in standard English?

Reviewer #2: Yes

Reviewer #3: (No Response)

6. Review Comments to the Author

Reviewer #2: --

Reviewer #3: (No Response)

7. PLOS authors have the option to publish the peer review history of their article (what does this mean?). If published, this will include your full peer review and any attached files.

Reviewer #2: No

Reviewer #3: No

---

## [Editor Report · Acceptance letter]

5 Jul 2021

PONE-D-21-06610R2 

Feasibility of motor imagery and effects of activating and relaxing practice on autonomic functions in healthy young adults: a randomised, controlled, assessor-blinded, pilot trial 

Dear Dr. Seebacher:

I'm pleased to inform you that your manuscript has been deemed suitable for publication in PLOS ONE. Congratulations! Your manuscript is now with our production department. 

Kind regards, 

on behalf of

Dr. Walid Kamal Abdelbasset 

Academic Editor

PLOS ONE